# Additive manufacturing of strong silica sand structures enabled by polyethyleneimine binder

Dustin B. Gilmer[1,2], Lu Han[1], Michelle L. Lehmann[1,2], Derek H. Siddel[3], Guang Yang [1], Azhad U. Chowdhury[1], Benjamin Doughty [1], Amy M. Elliott [3✉] & Tomonori Saito [1✉]

Binder Jet Additive Manufacturing (BJAM) is a versatile AM technique that can form parts from a variety of powdered materials including metals, ceramics, and polymers. BJAM utilizes inkjet printing to selectively bind these powder particles together to form complex geometries. Adoption of BJAM has been limited due to its inability to form strong green parts using conventional binders. We report the discovery of a versatile polyethyleneimine (PEI) binder for silica sand that doubled the flexural strength of parts to 6.28 MPa compared with that of the conventional binder, making it stronger than unreinforced concrete (~4.5 MPa) in flexural loading. Furthermore, we demonstrate that PEI in the printed parts can be reacted with ethyl cyanoacrylate through a secondary infiltration, resulting in an increase in flexural strength to 52.7 MPa. The strong printed parts coupled with the ability for sacrificial washout presents potential to revolutionize AM in various applications including construction and tooling.

---

[1] Chemical Sciences Division, Oak Ridge National Laboratory, Oak Ridge, TN, USA. [2] The Bredesen Center for Interdisciplinary Research and Graduate Education, University of Tennessee, Knoxville, TN, USA. [3] Manufacturing Science Division, Oak Ridge National Laboratory, Oak Ridge, TN, USA. ✉email: elliottam@ornl.gov; saitot@ornl.gov

Additive manufacturing (AM) enables the creation of unparalleled structures out of common materials[1–5] and provides a path to producing customized tools and dies for industrial manufacturing. Binder jet additive manufacturing (BJAM) is well suited for manufacturing these tools and dies due to its high production rates, low operator burden, high resolution, and ability to process low-cost feedstocks, such as sand[6]. BJAM is an AM process that uses a layer-by-layer polymer binding process to rapidly manufacture complex three-dimensional tools and dies at significantly higher throughput when compared with directed-energy binding AM methods[6,7]. Selective inkjetting is used to deliver the binding polymer into the layers of powder to bind the powdered materials together (Fig. 1)[6,8]. The process forms a porous preform (i.e., a green part) after curing at elevated temperature. The green part can then be post-processed in various ways, including sintering or polymer infiltration, to produce dense materials with higher mechanical strength[9]. Silica sand is an attractive powdered material that can be readily processed using BJAM, and it is especially suited for creating tools and dies due to its low coefficient of thermal expansion (CTE) coupled with its low cost[10–12]. However, the mechanical weakness of the green parts has been a major bottleneck, preventing the widespread use of BJAM of silica sand for tooling.

In BJAM, the binder is responsible for the mechanical strength of the green parts. The current state-of-the-art binder for silica sand is a no-bake or self-hardening binder system, based on a furfuryl alcohol polymerization or phenol-formaldehyde reaction. These binders impart relatively low mechanical strength, making the manufacturing of functional tooling challenging[13]. The low mechanical strength of the no-bake system can be linked to limited interfacial interactions between the polymer and the sand.

Thus, it is important to develop binders having strong interfacial interactions with the sand that will lead to strong green parts.

In the following study, we report a versatile binder, hyperbranched polyethyleneimine (PEI), for BJAM that has strong interfacial interactions due to the presence of its primary and secondary amine groups. The functionality enables the PEI binder to form strong green parts with silica sand, allows for reactive secondary infiltration, and exhibits washout functionality. PEI's hyperbranched structure also provides low viscosity, high solubility, and limited crystallinity, making it ideal for the piezo-electric drop-on-demand (DOD) inkjetting process used in BJAM[14–16]. These properties, along with a low molecular weight (~800 g/mol), allow PEI to be dispersed in a solvent mixture with a wide range of solid loadings, which enables fine control of the polymer content within the green part to tailor the strength for various applications. To understand the various factors that affect the green strength, extensive characterization on the PEI-sand interface was conducted and it revealed unique interfacial interactions between PEI and silica sand. The green strength can then be further increased by reactive secondary infiltration utilizing residual amine groups in the green part. Furthermore, the washout functionality of the PEI binder is demonstrated, which allows parts printed with PEI to serve as sacrificial tools for lightweight geometries and structures.

## Results and discussion

Each binder system must have its viscosity and surface tension adjusted with a solvent mixture for optimal droplet formation, and it also has to be balanced with the kinetics of the binder's penetration into the powder bed (Supplementary Table 1). The binder's penetration into the powder is important as it controls the rate and volume of binder incorporation into the green parts, which affects the print times and mechanical strength of the green parts. In this sense, conventional binders have limited formulation ranges, providing limited ability to alter print times and green strength[17], whereas the PEI binder has a wide range of possible formulations. For silica sand particles, 15 wt% PEI in 75:25 of water:1-propanol was identified as the optimal binder formulation (Supplementary Fig. 2). Testing a variety of PEI loadings in the ink optimized the printability, as well as binder penetration into the powder bed but did not elucidate the optimal volume of PEI in the part for maximal green strength. To find the maximum strength, an increasing volume of PEI was incorporated into the green part, resulting in a nearly linear increase in green strength (Fig. 2a). A sample with 5.5 wt% PEI net loading in the part, which was cured in an oven set at 180 °C for 2 hours (h), achieved a maximum green strength of 6.28 MPa (Fig. 2a). There was an observed decrease of green strength at 6 wt% PEI loading, which was likely due to a combination of factors such as fluid adsorption into the powder bed, solvent evaporation kinetics, and heat transference during curing. The achieved maximum flexural strength of 6.28 MPa exceeds the strength of printed silica sand parts using a commercial polymeric binder (ExOne binder), measured at 1.82 MPa, and that of the current furan binder system, at 3.60 MPa under optimized processing conditions (Fig. 2a). The flexural strength of the printed silica sand parts with PEI surpasses any other sand polymer mixtures, even that of unreinforced concrete, the most commonly used construction material on earth, which averages a flexural strength of 4.5 MPa[18,19]. This green part strength enables the creation of complex high-strength parts for tooling using low-cost silica sand at high throughputs, which can revolutionize tooling for the automotive, aerospace, and consumer industries.

The curing step in BJAM is critical in achieving high strength in green parts, as curing parts in ovens set at temperatures below

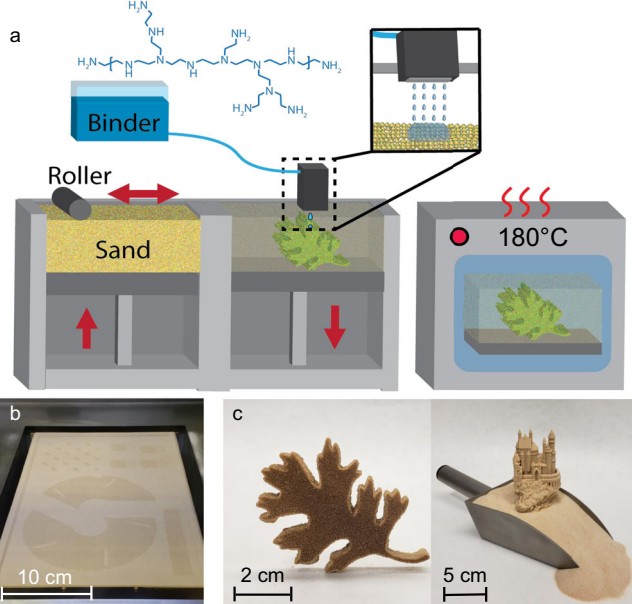

**Fig. 1 The BJAM process and representative printed parts. a** A diagram of the BJAM printer and curing process: The powder in the feed side of the printer, labeled "Sand," is moved to the build side by the counter-rotating roller, where the build side will be lowered after each layer of binder is deposited. The printed parts are cured at 180 °C. **b** Powder bed during printing showing the incorporation of the binder into the silica sand powder bed, which results in a darker pattern. **c** The complex self-standing sand features that are produced with BJAM showing (left) the leaf structure from the diagram in panel a and (right) a castle with intricate features printed upon a sand batch.

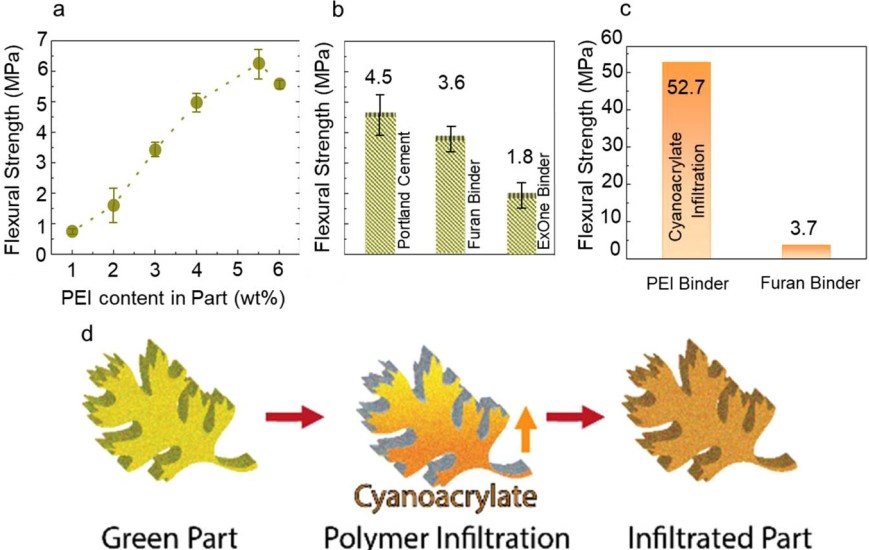

**Fig. 2 Mechanical properties of BJAM as printed and infiltrated parts comparing PEI with traditional binders and other sand composites. a** Flexural strength of green parts printed with 15 wt% PEI binder at varying saturations; every sample iteration was tested in triplicate and the standard deviation was taken to provide the error bars. **b** Comparison of PEI binder flexural strength with other commercial binders and cement; All samples were tested in triplicate and the standard deviation was used to calculate error bars. **c** Flexural strength of ethyl cyanoacrylate (ECA) infiltrated green parts comparing PEI binder vs. furan binder. **d** Infiltration of green parts with ECA.

180 °C impart significantly lower green strengths of PEI parts, such as 0.14 MPa with 2 h curing at 150 °C (Supplementary Table 3). To elucidate how curing affects the chemistry of PEI, carbon-13 nuclear magnetic resonance ($^{13}$C-NMR) and Fourier transform infrared spectroscopy (FT-IR) were performed on uncured PEI and PEI cured at 180 °C from 10 min to 5.5 h (Fig. 3). In the $^{13}$C-NMR spectra, carbon peaks associated with eight types of amine groups within the ethylene imine units can be seen before curing (Fig. 3b). These peaks consist of carbons associated with tertiary amines (1, 2, and 3), secondary amines (4,5, and 6), and primary amines (7 and 8) (Fig. 3b)[20,21]. The shifting and broadening of these peaks upon the curing process (Fig. 3b) suggest that the PEI is oxidizing during curing. The appearance of C=O and C=C functional groups at 1655 cm$^{-1}$ in FTIR (Fig. 3c) and at 167–158 ppm and 138–120 ppm in $^{13}$C-NMR (Fig. 3a) further confirms oxidation of PEI during the curing process[21,22]. These oxidation products, especially C=O, are known to function as hydrogen (H) acceptors, and the FT-IR spectra identify the increased presence of H-bonding at 3100–3400 cm$^{-1}$ with a longer curing time.

To support the linear spectroscopic measurements, surface-specific vibrational sum-frequency generation (SFG) spectroscopy was used to understand how PEI interacts with the silica surface to provide the aforementioned increase in mechanical strength (See supplementary discussion). First, we investigate the SFG spectrum of uncured PEI at a quartz interface plotted in Fig. 3d as a solid black trace. Here we find characteristic CH$_2$ stretches from the interfacial PEI consistent with expectation[23–26]. Upon curing, the sharp CH$_2$ vibrations disappear, and a new dominant peak appears at ~2900 cm$^{-1}$ (red trace). To identify the species formed upon curing, the NH groups on the PEI were deuterated (ND), which isotopically shifts the N/D stretching frequency of the uncured polymer out of this spectral window. The cured ND-sample spectrum (blue trace) does not show the intense peak at ~2900 cm$^{-1}$, indicating that the prominent band in the cured PEI sample is due to NH groups at the interface that are strongly shifted. Such a strong shift is descriptive of correspondingly strong H-bonds that form between the cured PEI and silica on curing. This change in H-bonding is accompanied by a

corresponding loss of the CH$_2$ vibrational signatures that were dominant in the uncured PEI spectra. Because SFG is primarily sensitive to functional groups aligned out of the interfacial plane[27–29], the results provide a clear indication of interfacial alignment. The uncured PEI layer is adsorbed to the silica interface in a conformation that allows only a few H-bonds, where the CH$_2$ groups are generally pointed out of the interfacial plane (Fig. 3e). Upon curing, the structure rearranges to maximize the number of H-bonds (evidenced by a strong increase in NH peak at ~2900 cm$^{-1}$), which forces the CH$_2$ groups in the PEI to contort into a corrugated-like structure to accommodate the H-bonding anchors (Fig. 3e). As such, the CH$_2$ groups are oriented more parallel to the interfacial plane and thus generate weaker SFG signals. This corrugated interfacial structure facilitates densely packed and tightly bound H-bonding interactions of >N–H⋯O–Si between the cured PEI and quartz that lead to the enhanced mechanical strengths described above.

The strength of silica sand green parts can be further increased through post-processing using secondary polymer infiltration (Fig. 2d). The infiltration is possible due to the inherent porosity of BJAM parts, which allows for low-viscosity polymers or monomers to absorb into the structure. Here ethyl cyanoacrylate (ECA) was used to infiltrate into the PEI green parts as it is expected to induce Michael-addition reactions with the amine groups on PEI. The secondary ECA infiltration resulted in an increase in the flexural strength by a factor of eight to 52.74 ± 2.18 MPa from 6.28 ± 0.48 MPa (Fig. 2c) and a decrease in porosity to 8.92% from 9.90% (See supplementary discussion). The ECA-infiltrated PEI part was compared with silica sand parts printed using the furan no-bake binder and infiltrated with ECA by the same method. The ECA-infiltrated furan sand part demonstrated an insignificant increase in flexural strength to 3.70 ± 0.24 MPa from 3.60 ± 0.45 MPa (Fig. 2c). The strength of the ECA-infiltrated PEI part indicates the occurrence of a reactive infiltration, such as a Michael-addition reaction between PEI and ECA, followed by chain polymerization of ECA. $^1$H-NMR was used to study the mixing of ECA and PEI (ECA to PEI ratio of 5:1) and confirmed the occurrence of Michael-addition reaction between ECA and the residual free amine groups of PEI within

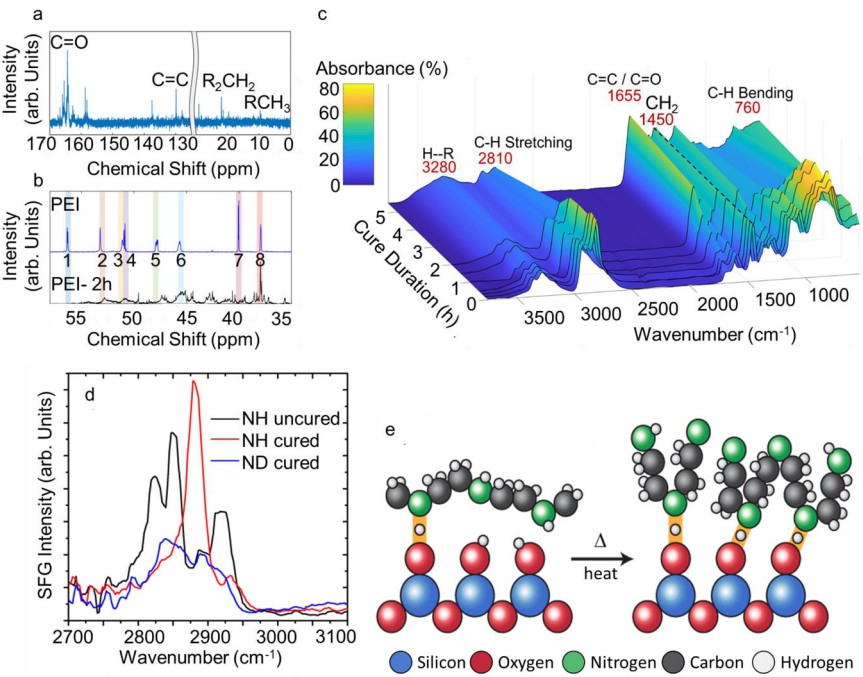

**Fig. 3 Chemical characterization of PEI binder. a** $^{13}$C NMR of PEI cured at 180 °C for 2 h between 30 and 0 ppm and 170 and 125 ppm. **b** Comparison of $^{13}$C NMR from PEI and PEI cured at 2 h between 60 and 30 ppm. The alteration in structures due to curing can be seen for carbons attached to eight types of amines: (1) N-CH$_2$-CH$_2$NH$_2$, (2) N-CH$_2$-CH$_2$NH, (3) N-CH$_2$-CH$_2$-N, (4) NH-CH$_2$-CH$_2$NH$_2$, (5) NH-CH$_2$-CH$_2$-NH, (6) NH-CH$_2$-CH$_2$N, (7) NH$_2$CH$_2$-CH$_2$N, and (8) NH$_2$CH$_2$-CH$_2$N. **c** FT-IR spectra of PEI as a function of curing time at 180 °C. **d** Sum frequency generation spectra from uncured PEI, cured PEI and isotopically labeled (ND) cured PEI. The molecular conformation at the PEI-quartz interface with uncured PEI is sketched in **e**, where few H-bonds are formed, and the PEI species lay on the surface with hydrophobic groups oriented away from the surface. Upon curing the molecular ordering changes to promote H-bonding with the surface: this drives PEI chains to pack tightly such that the CH$_2$ groups are pointed parallel to the surface plane, thereby suppressing sum-frequency generation signals from the -CH$_2$ groups. In panel e the colored balls represent Si (blue), O (red), N (green), C (black), and H (white), and solid orange lines denote H-bonding between the quartz surface and PEI.

the porous part (Fig. 4a)[30]. The resonances associated with the acrylate double bond in ECA disappeared, whereas the resonances associated with the PEI molecule are still present after the reaction. The reacted mixture showed the appearance of two new resonances. One resonance represents the formation of amine-acrylate linkage by the Michael-addition at 3.6 ppm, namely, a methylene bridge between the secondary or tertiary amines of PEI and the carbon atom originally from the C═C bond in ECA. The second new resonance around 1.4 ppm indicates the occurrence of anionic addition polymerization of ECA. Mechanistically, the two-electron withdrawing cyano and ester groups in ECA assist in triggering a nucleophilic attack by PEI on the double bond to initiate the chain polymerization (Fig. 4b). This linking reaction between the ECA and PEI leads to the high strength observed in the part. The high flexural strength of the infiltrated PEI-printed silica sand parts, ~53 MPa, surpasses construction materials such as masonry or brick-and-mortar structures, with the ability to support at least 300 times its own weight (Fig. 4c)[31,32]. The strength of the infiltrated PEI parts opens up structural applications similar to those of concrete and tooling applications such as autoclave tooling, which experiences high working pressures and temperatures[33]. Using this material for complex customized construction may create a revolution in customized structural applications, which could result in reducing environmental impacts and improving structural optimization on demand[34,35].

Washout tooling is a process in which the entire tool can be removed by simply washing it away with water or other solvents. The washout functionality enables the creation of tooling for the formation of complex, hollow composite structures[36]. For example, washout tooling can be used for the creation of ducting in the aerospace and defense sector and structurally graded cores

for composite sandwich panels, whose industries are projected to grow to a value of $7 billion by 2025[37,38]. The silica sand green parts formed with the PEI binder provide high strength but can also function as washout tools as they can be easily dissolved in water and washed out (Fig. 4d). The ability to dissolve the green part arises from the non-covalent hydrogen bonding that governs the interfacial interaction between the PEI binder and the silica surface. Water can disrupt the hydrogen bonding and redissolve the PEI in the parts, resulting in the removal of the binding material in the green part, as shown in Fig. 4d. All of the conventional toolings for creating hollow composites, such as inflatable bladders, plaster, or eutectic salts, have challenges centered around slow washout times, weak mechanical properties, and poor thermal properties[39–41]. Using BJAM to create tooling from silica sand with PEI can mitigate these issues, as the robust green parts solve a major bottleneck of producing viable complex preforms for washout tooling with BJAMtoday. Parts infiltrated with ECA also remain soluble in water (Fig. 4d), allowing them to be used for washout tooling. Moreover, the green parts and infiltrated parts exhibited low CTE of 14.73 μm/m°C and 20.81 μm/m°C, respectively, further reinforcing their suitability for the use in tools. It should be noted that ECA-infiltrated parts remain soluble only within ~48 h of the infiltration, and the composite will become insoluble after 48 h, forming a stable structure that can be used for permanent fixtures. The washable state of ECA-infiltrated parts is ideal for use in temporal preform applications, such as washout tooling, and the insoluble ECA infiltrated parts enable structurally robust permanent fixtures such as exterior and interior building materials.

Using the PEI binder for BJAM of silica sand enables the production of versatile green parts that provide high strengths,

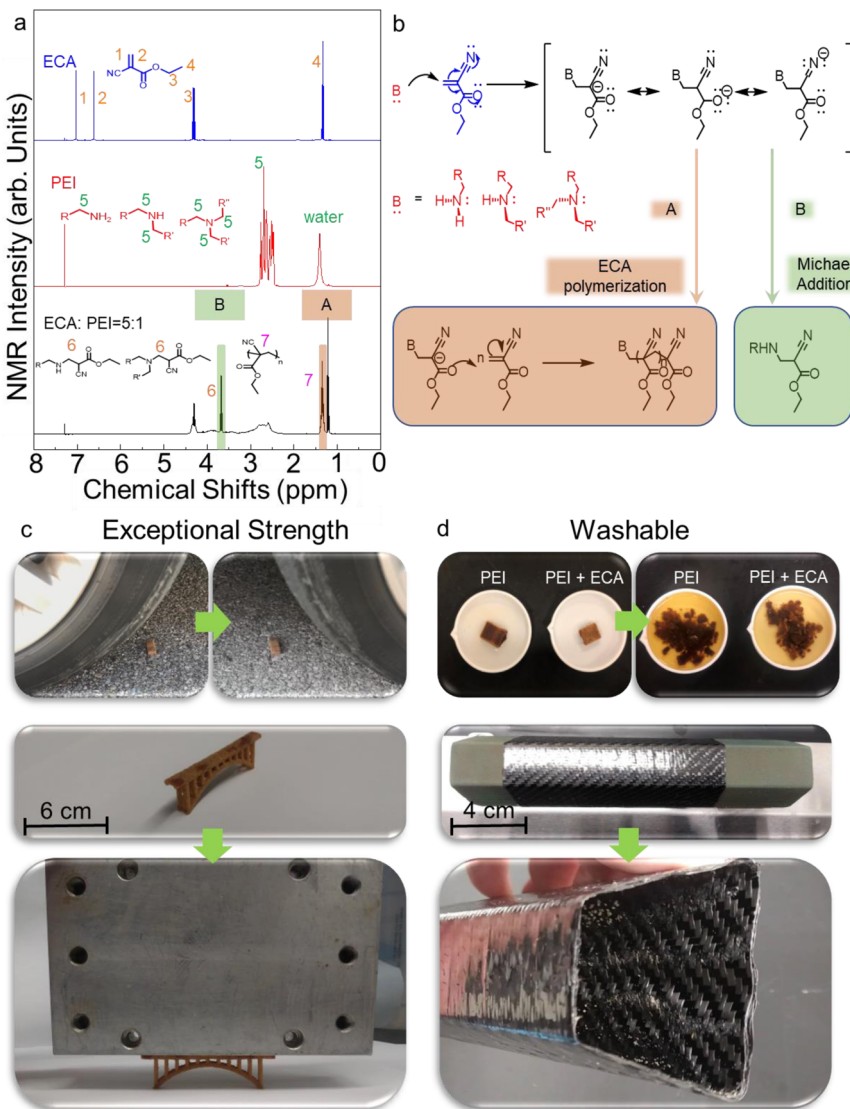

**Fig. 4 Reactive infiltration and demonstration of high strength and washout capability. a** [1]H NMR spectra of ECA (Blue), PEI (Red), and ECA: PEI = 5:1 (Black) in CDCl3 at room temperature. (1) and (2) Representative of C=C (3) Representative of CH2 (4) Representative of CH3 (5) Representative of all primary, secondary, and tertiary amine connected to CH2 (6) Representative of Michael Addition reaction (7) Representative of anionic addition polymerization of ECA. **b** Michael addition and ECA polymerization mechanism. **c** Demonstration of high-strength PEI + ECA sand part by the survival of a part run over by a car and supporting an aluminum block that is 300 times its weight on the 6.5 cm bridge. **d** Washout properties of PEI and PEI + ECA parts, and formation of advanced fiber composites by washout tooling.

sacrificial washout functionality, and reactive infiltration capability. The green strength of the silica sand parts is significantly enhanced to double that of the conventional binder by forming a corrugated nano-structure of interfacial H-bonding. The residual amine groups of PEI then allow for reactive ECA infiltration, achieving a further eightfold increase in the printed part's strength, exceeding any known polymer sand composites and building materials such as unreinforced concrete and masonry. The ability to create these strong components opens industrial tooling applications from autoclave tooling to metal stamping and provides the possibility to revolutionize the tooling industry using BJAM[33], where these tooling industries represented a combined value of $200.8 billion as of 2018[42]. It also opens industrial washout tooling applications toward the manufacturing of hollow composite structures for the aerospace and defense industries. The PEI binder enables these applications by creating versatile interfaces including corrugated interfacial H-bonding, reactive initiation sites for ECA polymerization, and washable

surfaces with water. The capability to form such versatile interfaces also provides an insight into the design of hybrid materials beyond BJAM. The findings in this study will significantly advance the AM technologies for industrial tooling, construction, and various other transformational applications.

## Methods

**Materials**. The utilized powder material is a commercial foundry silica sand (SiO2), with a mean diameter of 150 μm and bulk density of 2.8 g/ml, which was obtained from ExOne Corporation, USA. The PEI (branched polyethyleneimine (Mw ~ 800 g/mol and Mn ~ 600 g/mol) was obtained from Sigma Aldrich and used without further purification. 1-Propanol, ACS, 99.5 + %, and deionized water (H2O) were obtained from Sigma Aldrich and used as received. Iso-propanol (70% concentration) was obtained from McMaster-Carr and used as received.

**Printing**. An X1-lab binder jet printer from ExOne Corporation was utilized for BJAM and formed the specimens used in testing. The printer was equipped with a large nozzle printhead containing the SL-128 AA module from Fuji Films, which creates 80 pL drops and has 128 nozzles with a 50 μm diameter nozzle size. The printer utilizes a single counter-rotating roller for powder spreading and

compaction. When introducing a new binder to the X1-lab system, the previous binder must be cleared from the system utilizing isopropanol to remove the previous binder residue[43]. After flushing the system, the new binder is introduced to the printer fluid system. The new binder must be allowed to sit in the printer fluid system for several hours to permeate each individual jet/orifice in the print head to prevent uneven printing later in the process. The efficacy of each nozzle is then quantified using a sheet of colored paper onto which a test pattern is printed. This printed pattern allows for the quantification of all nozzle's operational status. Any jets that are not functioning can then be identified using the printer software and subsequently disengaged.

To load the selected powder into the printer, the build platform is lowered by 4 mm. This is the depth of the build plate, which is utilized to remove the print from the printer. The feed platform is then lowered to the distance required for the height of the build, dependent on the size of the part being printed. The powder is then poured into the feed side and over the build platform. It is required to pour excess of the powder and slightly mound it over in both the feed and build sides of the printer. The slight mounding of the powder ensures a flat surface after the roller is moved across the platform. The powder in the bed is then leveled using a counter-rotating bar. Print parameters can then be adjusted. The main print parameters include saturation, heat, and layer thickness, which need to be tuned depending on a binder, a powder, and the geometry of the print to achieve a successful print.

Saturation (S) is the measured value of how much binder volume is being added to the powder bed in the given geometry of a part, as described in Eq. (1). No powder bed is 100% dense, so saturation is the measure of void space in the part that is being filled with a binder.

$$S = \frac{V_{binder}}{V_{void}} = \frac{100,000 \times V_{drop}}{\left(1 - \frac{D}{100}\right) \times x \times y \times z} \quad (1)$$

$V_{binder}$ is the volume of the binder, and $V_{void}$ is the volume of pores in the powder bed, which is calculated based on the $D$, the powder packing density. In the case of the silica sand, the powder packing density was determined to be approximately 50% and this value was utilized in all calculations. $V_{drop}$ is the volume of one drop and $x$, $y$, $z$ is the spacing of the drops on the powder bed from each other in μm. To determine the optimal composition of PEI binder and silica sand, multiple saturations were printed and tested for green strength utilizing a three-point bend instrument. More information is available regarding this data and the compositions in the main text and supporting information.

Heat is emitted during the printing process to evaporate the excess solvent from the binder solution. During printing with the PEI binder, the heating was tuned based on the saturation, where higher saturations generally require higher heating power to evaporate the excess solvent introduced to the system.

The layer thickness is determined by the powder that is being printed. The silica sand powder from ExOne Corporation has a D50, which is the mean particle diameter of 100 μm. When printing with the ExOne X1-Lab system, it is suggested the layer height be double of the D50, so that it can encompass the majority of particles when the roller is moving the powder from the feed reservoir to the build reservoir. This layer height prevents particles that are larger than the D50 from disturbing the layer and causing issues during the build such as inconsistent layers and short spreading. Thus, the layer thickness for these prints was set at 200 μm for all prints.

**Binder fluid property**. To determine the printability of the fluids, the viscosity and surface tension were both measured and combined to form the Ohnesorge number, which is given in the supporting information.

Viscosity: The viscosity of four PEI binder solutions described in the supporting information were measured by Electromagnetically Spinning Viscometer (EMS) utilizing an EMS-1000 from Kyoto Electronics Manufacturing Co. The EMS method of measuring viscosity operates by placing the sample in a small test tube with an aluminum sphere inside, which is then placed inside of the instrument. The instrument contains two magnets attached to a rotor, which creates a rotating magnetic field. The rotating magnetic field will then induce eddy currents in the sphere which will cause the sphere to rotate. The torque applied to the sphere is proportional to the difference in the angular velocity of the magnetic field $\Omega_B$ and one of the spheres $\Omega_S$. The viscosity of the liquid is measured by creating a linear relationship between $(\Omega_B - \Omega_S)/\Omega_S$[44].

Surface Tension: The surface tension of four PEI binder solutions described in the supporting information were measured by the Wilhelmy plate technique in a custom-built instrument, in which a thin plate of glass oriented perpendicular to the interface is lowered into the liquid and the forces on the plate are measured. The surface tension is calculated from the Wilhelmy equation (Eq. 2).

$$\gamma = \frac{F}{l\cos(\theta)} \quad (2)$$

where $F$ is the force, $l$ is the wetting perimeter ($2w + 2d$) with $w$ being the plate width and $d$ being the plate thickness and $\theta$ is the contact angle between the liquid and the plate[45].

**Structural characterization**. [13]C NMR: [13]C NMR spectra were recorded for pure PEI and PEI heated in an oven for 2 h at 180 °C on a Bruker 400 MHz NMR spectrometer using $D_2O$ at 25 °C.

[1]H NMR: [1]H NMR spectra were recorded for ECA, PEI, and a mixture of ECA and PEI (ECA to PEI ratio of 5:1) on a Bruker 400 MHz NMR spectrometer using $CDCl_3$ at 25 °C.

FT-IR: FT-IR spectra were recorded on a Cary 600 Series FT-IR spectrometer (Agilent Technologies), with a scanning range of 4000–400 cm$^{-1}$. Analyzed samples include pure PEI and PEI heated in an oven under ambient atmosphere for 10, 20, and 30 min, and 1, 1.5, 2, and 5 h at 180 °C.

Sum Frequency Generation (SFG): A custom-built SFG spectrometer was used in this study[46,47]. The output of a regenerative amplifier system (Spectra-Physics Spitfire Ace) was divided into two paths using a beam splitter. One portion was sent to an optical parametric amplifier and difference frequency mixer to produce broadband infrared (IR) pulses, whereas the other portion was sent to a 4$f$-pulse shaper equipped with a 2D spatial light modulator to produce a narrowband near-infrared (NIR) optical pulse for up-conversion. The spectral bandwidth of the IR pulse was ~300 cm$^{-1}$ at full width at half-maximum (FWHM) when centered at ~2900 cm$^{-1}$. Individual polarizers were used to purify the IR and NIR polarizations before being rotated with zero-order half waveplates. A dichroic optic was used to combine the IR and NIR beams in a collinear geometry. The laser pulses were then focused on the sample at an angle of 60° with respect to the surface normal. The SFG signal was collected and re-collimated by an air-spaced achromatic doublet, passed through a waveplate-polarizer pair, filtered, and focused into an Acton SpectraPro SP-2300 spectrometer paired with a Pixis 256 CCD camera. The quartz-PEI interface was probed in reflection geometry. The sample was placed in a facedown geometry where the incident beam entered through the quartz side of the sample to avoid attenuation of the IR through the PEI polymer sample. The SSP (S-SFG, S-NIR, P-IR) polarization combination was used for all measurements. The raw SFG data were processed according to standard procedures detailed in our previous work[49]. The SFG samples were prepared by spin-coating 15 wt% PEI solution onto a quartz surface chemically equivalent to the silica sand surface. One coated sample was cured in the oven at 180 °C for 2 h, with the other sample tested without curing.

X-Ray Computed Tomography (XCT): The XCT measurements were performed using a Zeiss Metrotom 520 800 225 kV XCT instrument. Two different XCT scans were performed, one on a green sample and one on a sample after infiltration with ECA. The XCT uses scintillators that convert x-rays to visible photons, which are detected by a CCD camera. The choice of optics allows the user to select from several detector pixel resolutions. Data acquisition and reconstruction with the XCT setup were performed using the Zeiss Scout and Scan software v.11.

**Mechanical characterization**. Three-point bend: Mechanical testing was performed in triplicate on samples containing 1–6 wt% of PEI, Furan, and ExOne binders, as well as ECA, infiltrated samples of the PEI and Furan binders using a custom-made three-point bend instrument. The testing utilized MPIF Standard 15 or its equivalent to ASTM B312. This standard is conventionally used in the powder metallurgy industry, analogous to the process of BJAM. The 3-point bend instrument utilized a stepper motor that moved at a rate of 2.54 mm/min. The specimens were printed with the dimensions of 31.7 mm length, 12.7 mm width, and 6.35 mm height, following the standard specification. The flexural strength ($\sigma$) was determined with equation (Eq. 3).

$$\sigma = \frac{3FL}{2bd^2} \quad (3)$$

$F$ is the force at the fracture point, $L$ is the distance between two supports, $b$ and $d$ are the width and thickness of the tested sample, respectively. Every sample iteration was tested in triplicate and the standard deviation was taken to provide the error bars utilized within the figures.

**Thermal characterization**. Thermomechanical Analysis (TMA): Two samples of $5 \times 5$ cubes including a printed silica sand parts with 5.5 wt% PEI, and one infiltrated with ECA after printing were tested according to the ASTM E831. The TMA was performed to 185 °C at 2 °C per min and the cycle was repeated 5 times.

Differential Scanning Calorimetry (DSC): Glass transition temperature of PEI and cured PEI was measured using DSC (TA Instruments Q1000). The heating procedure was set to modulate ±1.00 °C every 60 s at the rate of 3 °C per min, from −160 to 90 °C.

**Post-processing**. Infiltration: The printed samples were infiltrated by placing them into a disposable container containing the ECA. The ECA is then absorbed into the sample through capillary forces, which draw the ECA monomer into the porous sample. It was allowed to absorb ECA for 1 h within a glove box to help to reduce the occurrence of ECA self-polymerization and then the sample was removed from the container and placed onto a Teflon dish to dry before removing from the glove box.

Carbon Fiber Layup: Casting molds having the dimensions of 5.72 cm width, 3.81 cm height, and 25.4 cm length were printed into a trapezoidal-

shaped bar by an ExOne M-Flex binder jet 3D printer. Carbon fiber-reinforced composite was then laid upon the surface of the casting mold and cured at 150 °C for 1.5 h.

Washout: Washout was performed on the bar-shaped specimens and casting molds after layups on their surface. Each sample was placed into a container, which was then filled with room temperature tap water. The samples began dissolving in contact with water. The sample was allowed to soak in the water for up to 1 h but usually required much less time to completely dissolve. In some cases, slight pressure was applied to utilize a metallic spatula for water to completely penetrate the samples to accelerate the washout process.

## Data availability

The data that support the findings of this study are available from the corresponding authors upon reasonable request.

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

## Acknowledgements

The research was sponsored by the US Department of Energy, Office of Energy Efficiency and Renewable Energy, Advanced Manufacturing Office. B.D. and A.U.C. performed sum-frequency generation measurements and analysis and were supported by the US Department of Energy, Office of Science, Basic Energy Sciences, Chemical Sciences, Geosciences, and Biosciences Division. The authors would like to thank ExOne and specifically Dan Brunermer, Huayun Yu, and Rick Lucas for their input and advice on this research. The authors would also like to thank Harry Meyer, who assisted with x-ray photoelectron spectroscopy measurement, and Alan Druschitz from Virginia Tech for his assistance with casting research. The authors would also like to thank Jackson Wilt for his assistance with figure formation. In addition, the author would like to thank Andres E Marquez Rossy and Quinn A Campbell for their assistance in collecting and

collating the XRT data. The author is grateful for a fellowship from the Bredesen Center for Interdisciplinary Graduate Education.

## Author contributions

T.S. conceived the idea and led the research with A.M.E. D.B.G., and D.H.S., and M.L.L. formulated and printed the materials and specimens. D.B.G., M.L.L., G.Y., and L.H. performed characterization of the materials. B.D. and A.U.C. conducted SFG experiments and analysis. D.B.G, L.H., and T.S. wrote the manuscript. All authors read and revised the manuscript.

## Competing interests

The authors declare no competing interests.
