## [Peer Review File · Nature Communications]

Title: Additive Manufacturing of Strong Silica Sand Structures enabled by Polyethyleneimine BinderREVIEWER COMMENTS

Reviewer #1 (Remarks to the Author):

This paper describes a very promising advancement in binders for binder. The results show a substantial improvement and significant information is provided regarding the chemistry of the binder and its methods of operation. Additional detail is needed about the processing implications and test conditions.

Specific Suggestions:

- Abstract compares strength to concrete without specifying the strength type (tensile, flexural, compressive). Please revise to clarify.
- Figure 2 is confusing because it refers integrally to cyanoacrylate infiltration which is not mentioned until several pages later. Consider revising the flow or figure.
- Line 93: "5.5 wt% PEI loading": Need to clarify whether this is net loading in the part after printing or if this was the concentration in the binder
- What values of saturation were used for printing? This is not stated even though it is defined
- The authors report that the PEI binder requires curing at 180 C, but the conditions are not fully specified. Is this the oven temperature that is placed in or the temperature measured in the powder? If it is just the oven temperature, what is the side of the build box?
- The curing time for a large part (which is of great interest for most of the applications identified for these silica sand materials) could be very long due to the slow heating of powdered materials. This issue should be addressed in the article.
- The porosity of the final part can be very critical for some mold/tooling applications. This should be reported both with PEI alone and after infiltration with ECA
- the method of measuring the binder fluid properties is reported, but the resulting values are not.

Reviewer #2 (Remarks to the Author):

This is a paper of great interest to the scientific community. The authors investigated a novel binder chemistry to enhance the mechanical performance of 3D printed silica sand. A useful application of this system targets the composite tooling field.

The paper contains well detailed experimentation and background sections, and the supplemental material display robust information.

I support the publication of the present paper. However, I would like to make the following suggestion:

*Probably the CTE analysis of the system would be a great addition to the paper. How the inclusion of the PEI affected the CTE of the printed sand? Maravola et al. "Additive Manufacturing 25 (2019) 59-63 have published some data on the CTE of printed sand.

Reviewer #3 (Remarks to the Author):

The article Additive Manufacturing of Strong Silica Sand Structures enabled by Polyethyleneimine Binder concerned research on the preparation of green samples based on silica sand, polyethyleneimine as binder and ethyl cyanoacrylate as infiltration agent. The effects of application PEI and infiltration agent (ECA) on the flexural strength were determined. This manuscript make a major contribution to the development of additive manufacturing techniques. The presented results are novel and especially distinguished. The article is very interesting but while reading it I found a few issues that are not completely clear to me. Here they are:

- 1) Page 5, The value of flexural strength of samples based on furan phenol are different in the text (line 100 and line 175 Page 8) and in the Fig. 2.6. Why?
- 2) Were the density and porosity of prepared green samples measured? Were the differences between values of these parameters for samples containing only PEI and for samples containing PEI and ECA?
- 3) What is the bulk density of silica sand, what was the measuring procedure?
- 4) In Fig. 4. d, the samples signatures are confusing. In whole manuscript the abbreviation for ethyl cyanoacrylate is ECA and in this Figure is CA. Where did this difference come from?

REVIEWER COMMENTS

Reviewer #1 (Remarks to the Author):

This paper describes a very promising advancement in binders for binder. The results show a substantial improvement and significant information is provided regarding the chemistry of the binder and its methods of operation. Additional detail is needed about the processing implications and test conditions.

We would like to thank the reviewer for the kind regards and suggestions. We have addressed the following comments and have carefully revised the manuscript.

Specific Suggestions:

- Abstract compares strength to concrete without specifying the strength type (tensile, flexural, compressive). Please revise to clarify.

We would like to thank the reviewer for the comment. In the abstract page 1 line 27, we added the following addition in red to clarify the comparison of strength between our samples and concrete.

We report the discovery of a versatile polyethyleneimine (PEI) binder for silica sand that doubled the flexural strength of parts to 6.28 MPa compared with that of the conventional binder, making it stronger than unreinforced concrete (~4.5 MPa) in flexural loading.

- Figure 2 is confusing because it refers integrally to cyanoacrylate infiltration which is not mentioned until several pages later. Consider revising the flow or figure.

We thank the reviewer for pointing this out and we agree that this figure is slightly confusing due to the infiltration information not appearing till later in the paper. Considering the order of occurrence in the text, we have rearranged former Figures 2b and 2c to be above former Figure 2a to make them Figure 2a and Figure 2b and changed the former Figure 2a to Figure 2c. The former Figure 2a (cartoon) was not discussed until later and presenting as Figure 2c should flow better. This data of ethyl cyanoacrylate infiltration is intimately linked to the green strength data, and it is important to present the mechanical strength figure side-by-side. Thus, we believe it is the best that Figure 2a and Figure 2b (former Figure 2b and Figure 2c) are presented together, followed by a cartoon (Figure 2c), where the color of cartoon corresponds to those Figure 2a and 2b. The modified figure is shown below for the convenience of the reviewer.

- Line 93: "5.5 wt% PEI loading": Need to clarify whether this is net loading in the part after printing or if this was the concentration in the binder

We have clarified on page 5 line 93 as the following in red.

A part with 5.5 wt% PEI **net loading** in the part cured **in an oven set at 180°C** for 2 h achieved an outstanding maximum green strength of 6.28 MPa (Fig. 2a).

- What values of saturation were used for printing? This is not stated even though it is defined

We thank the reviewer for the comment. To clarify this experimental detail, a section was added to the supporting information by inserting an additional figure showing the increase in saturation correlating to wt%.

The modified supporting information is shown below.

3. **Optimal Binder Saturation/ Volume**

In addition to optimizing the binder formulation with a variety of PEI loadings in the ink, the optimal binder saturation in the printing process was evaluated to produce parts with the maximal green strength. To achieve the maximum green strength, the binder saturation was systematically increased, which is proportional to the increased wt% of the polymer in the part (Figure 2a and discussion in the main text). The flexural strength as a function

of saturation with correlation to the PEI wt% in the part is described in Fig. S3 and Table S2.

Figure S3. Flexural strength of green parts printed with 15 wt% PEI binder at varying saturations, and correlation to PEI wt% in part

Table S2. Effect of saturation on PEI content in part and flexural strength.

Printing Saturation	PEI content in part (wt%)	Flexural Strength (MPa)
30% Saturation	1.50	3.42
60% Saturation	3.00	5.00
90% Saturation	4.51	6.23
120% Saturation	6.01	5.58

- The authors report that the PEI binder requires curing at 180 C, but the conditions are not fully specified. Is this the oven temperature that is placed in or the temperature measured in the powder? If it is just the oven temperature, what is the side of the build box?

We thank the reviewer for this comment. The following sections in red were added to the manuscript for the clarification on page 5 line 93 and on page 6 and line 112-113. When the build box enters the oven, it will be at room temperature, the side of the build box will equilibrate to the temperature of the oven over the course of heating and thus the part will ultimately reach 180°C. The majority of testing was completed on small samples printed in an X1-Lab printer, and adverse effect by gradient thermal profile is not expected since there was minimal material in the oven to be heated.

A part with 5.5 wt% PEI **net loading in the part** cured **in an oven set at** 180°C for 2 h achieved an outstanding maximum green strength of 6.28 MPa (Fig. 2b).

The curing step in BJAM is critical in achieving high strength in green parts, as curing **parts in an oven set at** temperatures below 180°C impart significantly lower green strengths of PEI parts, such as 0.14 MPa with 2 h curing at 150°C (supplementary table).

- The curing time for a large part (which is of great interest for most of the applications identified for these silica sand materials) could be very long due to the slow heating of powdered materials. This issue should be addressed in the article.

We agree with the reviewer that curing time needs to be adjusted for a large part. To clarify curing adjustment needs, the following section was added to the supporting material on page 6.

The increased T_g caused by longer curing times (Fig. S4) along with the enhanced interfacial interactions (Figure 3 and discussion in the main text) indicates that the thermal curing of PEI is integral in achieving high strengths due to the chemical and conformational change. It is important to note that as the size of parts are increased through the printing on larger systems, the duration of the thermal curing cycle should be increased to account for delay in reaching the curing temperature in the part. For example, significantly large parts may need to be cured for overnight (e.g., 18 h) to make sure the chemical and conformational changes taking place uniformly throughout the part.

- The porosity of the final part can be very critical for some mold/tooling applications. This should be reported both with PEI alone and after infiltration with ECA

We thank the reviewer for this enquiry. We have quantified the porosity of the green samples and the infiltrated samples by utilizing a Zeiss X-Ray CT (XRT). The analysis is now included in the main text on page 8 line 172-173 and in the supporting material on page 8. The analysis by XRT allowed us to quantify the porosity of the green sample to be 9.90 %. After infiltration, the materials porosity decreased to 8.92 % showing that close to 1% of the total porosity was filled in with the ethyl cyanoacrylate in-situ polymerization.

The following addition is made in the main text on page 8 line 172-173.

The secondary ECA infiltration resulted in an increase in the flexural strength by a factor of eight to 52.74 ± 2.18 MPa from 6.28 ± 0.48 MPa (Fig. 2b) **and a decrease in porosity to 8.92% from 9.90% (see supplementary information for more detail).**

The following addition is made in the supporting material on page 8.

8. X-Ray CT scanning and Porosity Report

The porosity of the printed and infiltrated samples were analyzed by a Zeiss Metrotom 800 X-Ray CT (XRT) system. This XRT system reports the density variations within samples, that can differentiate between areas of high and low density or in the areas where voids are present (Video 1). This can then be used to describe the porosity of a given sample in bulk terms. XRT determined that the green samples had a porosity of 9.90 % and 8.92 % after ECA infiltration, indicating that ~1% of the total porosity was filled in with the ECA in-situ polymerization.

- the method of measuring the binder fluid properties is reported, but the resulting values are not.

We thank the reviewer for the comment. We described the fluid properties in section 1 in the supporting information on page 1 and 2. To ensure that readers are aware of this data, we have added the section in red below to the manuscript on page 4 line 83.

Each binder system must have its viscosity and surface tension adjusted with a solvent mixture for optimal droplet formation, and it also has to be balanced with the kinetics of the binder's penetration into the powder bed (see supplementary material for fluid properties).

Reviewer #2 (Remarks to the Author):

This is a paper of great interest to the scientific community. The authors investigated a novel binder chemistry to enhance the mechanical performance of 3D printed silica sand. A useful application of this system targets the composite tooling field.

The paper contains well detailed experimentation and background sections, and the supplemental material display robust information.

I support the publication of the present paper. However, I would like to make the following suggestion:

We would like to thank the reviewer for the kind regards on this work and the suggested study to improve the manuscript. We have addressed the following comment and have carefully revised the manuscript accordingly.

*Probably the CTE analysis of the system would be a great addition to the paper. How the inclusion of the PEI affected the CTE of the printed sand? Maravola et al. "Additive Manufacturing 25 (2019) 59-63 have published some data on the CTE of printed sand.

We would like to thank the reviewer for this suggested analysis. We agree that the CTE would be a great addition to the paper. Thus, we have conducted CTE analysis of a green part printed with PEI as well as a part infiltrated with ethyl cyanoacrylate using Thermomechanical Analysis. We have added a sentence in line 221 and 224 on page 11 and copied below for the

convenience of the reviewer. We have also added a section in the supporting material with this information in more detail, which is also included below. The testing procedure was identical to Maravola et al. "Additive Manufacturing 25 (2019) 59-63 and is included in the methods section with citation.

In line 222-223 on page 11,

Parts infiltrated with ECA also remain soluble in water (Fig. 4d), allowing them to be used for washout tooling. Moreover, the green parts and infiltrated parts exhibited low CTE of 14.73 $\mu\text{m}/\text{m}^\circ\text{C}$ and 20.81 $\mu\text{m}/\text{m}^\circ\text{C}$, respectively, further reinforcing their suitability for the use in tools.

In supporting information on pages 8 and 9,

9. Thermomechanical Analysis

A thermomechanical analysis was performed on the printed and infiltrated materials to elucidate the Coefficient of Thermal Expansion (CTE), and the results are shown in Table S3. Here, the plain printed silica sand with PEI yielded an average CTE of 14.73 $\mu\text{m}/\text{m}^\circ\text{C}$ at 185°C. Once infiltrated with ECA, the CTE increased to 20.81 $\mu\text{m}/\text{m}^\circ\text{C}$ at 185°C, which is expected from higher CTE of polyECA, 96.10 $\mu\text{m}/\text{m}^\circ\text{C}$ at 185°C. These low CTE values makes them well suited for uses in tooling applications such as composite layups.

Table S3. CTE of silica sand samples printed with PEI and CTE of parts infiltrated with ECA

Material	CTE ($\mu\text{m}/\text{m}^\circ\text{C}$) at 185 °C
Silica Sand Printed with PEI	14.73
Infiltrated with ECA	20.81
PolyECA	96.10

Reviewer #3 (Remarks to the Author):

The article Additive Manufacturing of Strong Silica Sand Structures enabled by Polyethyleneimine Binder concerned research on the preparation of green samples based on silica sand, polyethyleneimine as binder and ethyl cyanoacrylate as infiltration agent. The effects of application PEI and infiltration agent (ECA) on the flexural strength were determined. This manuscript makes a major contribution to the development of additive manufacturing techniques. The presented results are novel and especially distinguished. The article is very interesting but while reading it I found a few issues that are not completely clear to me. Here they are:

We would like to thank the reviewer for his or her thorough review and kind regards on this study. We have addressed the following comments and have carefully revised the manuscript accordingly.

1) Page 5, The value of flexural strength of samples based on furan phenol are different in the text (line 100 and line 175 Page 8) and in the Fig. 2.6. Why?

We thank the reviewer for the comment. In the referenced specific lines, the strength is correct as it is listed as 3.6 MPa in line 100 and in line 176 prior to infiltration. After infiltration the Furan samples strength is increased to 3.7 MPa as indicated in line 175 on page 8. Figure 2b was incorrect in its listing of both strengths at 3.8 but has been altered to reflect the correct values listed in the manuscript and is shown in its correct version below.

2) Were the density and porosity of prepared green samples measured? Were the differences between values of these parameters for samples containing only PEI and for samples containing PEI and ECA?

We thank the reviewer for this enquiry. Originally the density and porosity of the green samples and the infiltrated samples were not experimentally tested. As the other reviewer (reviewer 1) also enquired about the density/porosity of the samples, we had samples tested in a Zeiss Metrotom 800 X-Ray CT system. The analysis by X-Ray CT system shows that the green samples had 9.90 % porosity and after infiltration the material decreased in porosity to 8.92 %.

The detail was added in the main text on page 8 line 172-173 and supporting material page 8 as described in the response to the reviewer 2.

3) What is the bulk density of silica sand, what was the measuring procedure?

We thank the reviewer for this comment. The bulk density of silica sand is 2.8 g/ml. These values were included in the materials section in the main manuscript on page 12 line 250. This value was obtained from literature and measured by Archimedes method.

The utilized powder material is a commercial foundry silica sand (SiO_2), with a mean diameter of 150 μm and bulk density of 2.8 g/ml, which was obtained from ExOne Corporation, USA.

4) In Fig. 4. d, the samples signatures are confusing. In whole manuscript the abbreviation for ethyl cyanoacrylate is ECA and in this Figure is CA. Where did this difference come from?

We thank the reviewer for pointing out this oversight. In Fig. 4D, the sample signature should have read as ECA and not CA and has been altered to reflect the correct usage from the manuscript.

REVIEWERS' COMMENTS

Reviewer #1 (Remarks to the Author):

The changes address the concerns raised in the original manuscript. This is a great work.

Reviewer #2 (Remarks to the Author):

The authors have resolved all the corrections. The paper should be accepted for publication.

Reviewer #3 (Remarks to the Author):

I approve the revised manuscript and I have no more comments.